# Current Status and Future Perspectives of Optic Nerve Imaging in Glaucoma

**DOI:** 10.3390/jcm13071966

**Published:** 2024-03-28

**Authors:** Claudia Lommatzsch, Christian van Oterendorp

**Affiliations:** 1Department of Ophthalmology, St. Franziskus Hospital, Hohenzollernring 74, 48145 Muenster, Germany; 2Department of Ophthalmology, University of Luebeck, 23562 Luebeck, Germany; 3Department of Ophthalmology, University Medical Center Göttingen, 37075 Göttingen, Germany; christian.oterendorp@med.uni-goettingen.de

**Keywords:** glaucoma, optic nerve, optical coherence tomography, biomarker, optical coherence tomography angiography

## Abstract

Being the primary site of degeneration, the optic nerve has always been the focus of structural glaucoma assessment. The technical advancements, mainly of optical coherence tomography (OCT), now allow for a very precise quantification of the optic nerve head and peripapillary retina morphology. By far the most commonly used structural optic nerve parameter is the thickness of the parapapillary retinal nerve fiber, which has great clinical utility but also suffers from significant limitations, mainly in advanced glaucoma. Emerging novel imaging technologies, such as OCT angiography, polarization-sensitive or visible-light OCT and adaptive optics, offer new biomarkers that have the potential to significantly improve structural glaucoma diagnostics. Another great potential lies in the processing of the data already available. Artificial intelligence does not only help increase the reliability of current biomarkers but can also integrate data from various imaging modalities and other clinical measures to increase diagnostic accuracy. And it can, in a more efficient way, draw information from available datasets, such as an OCT scan, compared to the current concept of biomarkers, which only use a fraction of the whole dataset.

## 1. Introduction

Glaucoma, as one of the most common eye diseases, is identified by the gradual degeneration of retinal ganglion cells (RGCs), leading to a thinning of the retinal nerve fiber layer (RNFL). This layer consists of approximately 1.2 million unmyelinated axons of RGCs that converge there to leave the eye through Bruch’s membrane opening (BMO) and the scleral canal to constitutethe optic nerve. RGCs are particularly vulnerable to damage caused by elevated intraocular pressure, a characteristic feature of glaucoma. The loss of RGCs leads to specific spatial patterns of visual field loss that align with the bundle patterns of RGC axons. As most cases of glaucoma are completely asymptomatic over a very long period of time and the resulting symptoms are generally only noticed subjectively at a late stage, the disease is still the most common cause of visual impairment and irreversible blindness [1], with a high relevance as a health problem.

Assessing the anatomical loss in the RNFL is crucial for the diagnosis and monitoring of glaucoma. Optical coherence tomography (OCT) has emerged as a groundbreaking technology in ophthalmology, particularly in the field of glaucoma diagnostics, largely replacing the measurements of the Heidelberg Retina Tomograph (HRT) and the Scanning Laser Polarimeter (GDx).

With its ability to provide high-resolution cross-sectional images of the RNFL at high scanning speeds and good image quality, OCT plays a pivotal role in the early detection and monitoring of glaucoma. These advancements have also facilitated the visualization of critical structures deep within the optic nerve head (ONH), such as the lamina cribrosa (LC). This article explores the latest developments and capabilities of OCT in the field of glaucoma diagnostics concerning the optic nerve.

## 2. The Current Standard

### 2.1. Optical Coherence Tomography

The development of OCT technology has undergone significant advancements over the years.

Time domain (TD)-OCT represented the first generation of OCT systems. It relied on mechanical scanning of the mirror to measure the depth of the scan in the A-scan mode. However, TD-OCT systems had limited speeds and resolutions, restricting real-time imaging. The mechanical scanning led to longer scan times and limited clinical applications. Spectral-domain OCT (SD-OCT) introduced a crucial improvement by replacing mechanical scanning with spectral analysis. This allowed for higher imaging speeds and resolutions. SD-OCT systems became faster and more efficient, expanding clinical applications. The increased sensitivity led to improved image quality, enabling detailed examinations of retinal structures. The reproducibility of the measurements has been analyzed and proven in several studies [2,3]. Swept-source OCT (SS-OCT) is the latest development in OCT technology. It utilizes a tunable laser as the light source, allowing for further enhancement of scanning speed. SS-OCT provides improved depth resolution and enables imaging of deeper structures, such as the choroid. This technology has proven particularly beneficial in imaging the optic nerve head and LC.

OCT allows for a detailed visualization of the optic nerve head, including the BMO, the neuroretinal rim (often referred to as the minimum rim width—MRW), and the peripapillary nerve fiber layer (RNFL). The BMO functions as a reference point, indicating the transition from the retina to the optic nerve head (ONH). It aids in configuring OCT for the quantitative assessment of ONH structure, glaucoma diagnosis, and monitoring changes over time.

The representation of the RNFL and BMO-MRW is often in the form of an RNFL chart employing Garway–Heath sectors. This graph provides a detailed visual representation of the nerve fiber layer and NNR, considering specific sectors. The Garway–Heath sectors are six defined areas around the optic nerve head, represented in the RNFL chart. They include the temporal upper, temporal lower, nasal upper, nasal lower, temporal, and nasal quadrants. BMO-MRW is presented in the same way by some manufacturers. 

When analyzing and assessing the image results, it is important to consider not only the Garway–Heath sectors, but also the RNFL and MRW curves. Minor focal areas of impairment might appear as green since the instrument averages the thickness within a specific sector. Although there is localized loss, the overall average frequently falls within the normal range. Consequently, the entire printout appears green, creating the perception of an absence of damage. This constellation is known as “green disease”. On the other hand, there is also the case of “red disease”, when the OCT mistakenly indicates that something is abnormal (red), when there is no damage.

Some devices quantitatively determine other ONH-specific measurement parameters such as the optic nerve head area, the rim and excavation volume and the excavation. The optic disc area is usually determined by segmenting the visible area of the optic disc on the OCT images and not usually determined directly by detecting the Bruch membrane opening. The OCT system software identifies the boundaries of the optic disc, including the cup and disc, and then calculates the area within these boundaries. This process requires precise segmentation of the different tissue layers in the optic disc to enable accurate measurements. The quantitative data, including the optic disc area, are calculated by the OCT system and displayed in numerical form. These data can be used by the ophthalmologist to track changes in the optic disc over time and make a diagnosis.

Modern OCT technologies not only provide high-resolution cross-sectional images but also allow for the 3D reconstruction of the optic nerve head. This opens up new perspectives for a more comprehensive analysis and evaluation of structures related to glaucoma (see also Section 3.3).

While various OCT devices from different manufacturers may have specific features and printout formats, it is essential to focus on understanding the general principles of OCT interpretation that can be applied across different devices. The key features and parameters crucial for glaucoma assessment include retinal nerve fiber layer thickness and ganglion cell complex analysis. By understanding the fundamental aspects, ophthalmologists can effectively utilize OCT technology for glaucoma diagnosis and management, regardless of the specific device they use in their practice. It is important to be familiar with the normal appearance of the optic nerve head, the retinal nerve fiber layer, and the ganglion cell complex on OCT scans, as well as recognize patterns of abnormality indicative of glaucomatous damage. Additionally, understanding the limitations and potential artifacts associated with OCT imaging is crucial for accurate interpretation and clinical decision making.

#### 2.1.1. Current Structural Biomarkers Utilizing OCT

Structural biomarkers in the context of OCT in glaucoma refer to measurable anatomical features and changes in the eye’s tissue structures that can be captured through OCT. These measurements enable the identification of tissue changes and play a crucial role in diagnosing and monitoring of the disease.


**Retinal nerve fiber layer (RNFL) thickness:**


Thinning of the RNFL is a hallmark of glaucoma. OCT quantifies RNFL thickness, particularly in the peripapillary region, enabling precise assessment. Typically, the RNFL initially thins temporally, either inferiorly and/or superiorly, before progressing to hemispheric or diffuse further atrophy. Therefore, parameters such as the global average circumpapillary RNFL thickness and the RNFL thickness in the superotemporal and inferotemporal sectors are frequently used due to their higher diagnostic precision. The inferior RNFL is especially susceptible to glaucomatous damage and subsequent localized visual field defects, attributed to a reduced number of RNFL bundles originating from the inferior part of the macula compared to the superior macula. It is crucial to acknowledge the variability in the localization of RNFL loss, as not all glaucoma cases exhibit the same pattern. Examining these specific regions using OCT allows health care professionals to identify glaucomatous changes at an early stage, facilitating a more precise diagnosis. A recent study revealed that the proportion of eyes exhibiting abnormal average RNFL thickness four years before detecting visual field defects was 35%, and 19% of eyes showed abnormalities eight years prior [4]. Structural changes identified through OCT correlate directly with functional defects in the visual field associated with glaucoma. A decrease in RNFL thickness often precedes the occurrence of visual field defects, and the analysis of these structural changes can indicate early signs of glaucoma. The structure–function relationship provides a comprehensive assessment of the impact of glaucoma, contributing to a more precise diagnosis, better monitoring of the disease progression, and personalized adjustment of treatment accordingly.


**The minimum rim width (MRW)**


The width of the neuroretinal rim margin (NRR) can be depicted using OCT at defined measurement points as the minimum distance between the edge of the Bruch membrane opening and the internal limiting membrane (ILM). The BMO is determined either from a papillary volume scan or from a centered radial star scan with 24 scans and 48 measuring points. In a healthy eye, the thickness profile of BMO-MRW should show a slight double hillock. The mean area of the BMO is calculated for the non-high myopic normal population with 1.89 mm [2,5].

In the context of glaucoma disease, focal or diffuse thinning of the neuroretinal rim regularly occurs with increasing excavation. The atrophy of the NRR is typically initially reflected in a focal emphasis on thinning of BMO-MRW, with an initial emphasis temporally, superiorly and inferiorly. With further progression, the loss of the NRR becomes temporally pronounced, eventually resulting in a circular, total atrophy and significantly reduced MRW.

Since the adoption of BMO-MRW as the measure for assessing the NRR, research has delved into its effectiveness in detecting glaucoma, evaluating structure–function relationships, and pinpointing structural deterioration indicative of glaucoma progression. Studies indicate that BMO-MRW and RNFL (examples in Figure 1) thickness demonstrated similar discriminatory abilities between preperimetric and perimetric glaucoma eyes and normal subjects [6,7].

The findings regarding the identification of structural deterioration indicative of glaucoma progression suggest that RNFL thickness is more likely than BMO-MRW to exhibit a declining trend over time. Globally and within inferior and superonasal sectors, the longitudinal signal-to-noise ratios for RNFL were more negative than those of BMO-MRW, meaning that the information within longitudinal RNFL thickness measurements is less noisy compared with that of BMO-MRW [8].

#### 2.1.2. Particularities and Pitfalls in the Evaluation of OCT Images 

OCT poses continued challenges. Understanding the limitations and potential sources of error of the individual tests makes it possible to recognize false-positive or -negative results. In the following, some individual possible pitfalls of the respective measured structure will be discussed.

##### Evaluating OCT Data with a Normative Database

The acquired OCT data are compared with a normative database containing reference data from a healthy population. This comparison helps determine how individual measurements compare to the norm. Statistical analysis is performed to determine whether the measured values fall within the normal range or show deviations from the norm. 

Based on the statistical analysis, reports are created that present the results in an understandable way. These reports can help the ophthalmologist to assess the health of the eye and take any necessary measures as well as enable early detection of eye diseases or other abnormalities even before clinical symptoms appear.

The accuracy of the evaluation depends on the quality and representativeness of the normative database. There are sometimes considerable differences between the individual devices. What they all have in common is that limited pediatric data are stored. To the best of our knowledge, there is a scarcity of normative OCT data available for children, and further research is needed to establish comprehensive normative databases for pediatric populations. The data on ethnic groups, age and refraction, for example, are heterogeneous, potentially leading to incorrect conclusions if the database lacks diversity. However, the use of a normative database in the evaluation of OCT data offers a valuable opportunity for accurate diagnosis and monitoring of ocular disease when used judiciously and with consideration of its limitations.

##### OCT Imaging Quality 

In addition, good and reliable evaluation of OCT images requires good image quality. However, this can be negatively influenced by various factors, impacting the accuracy of interpretation. Opacities in the vitreous for example can result in blurry, hazy, or unclear images by reduction in light transmittance. This can also be caused, for example, by an unstable tear film and media opacities such as cataract. Furthermore, inadequate patient fixation can lead to motion artifacts. Even slight eye movements can significantly impact image quality, causing blurring during movement and disrupting tissue segmentation. Automatic segmentation of OCT images is prone to errors, especially in the presence of irregularities or pathological changes in tissue structures such as epiretinal membrane and retinoschisis. These errors can lead to incorrect measurements, affecting the diagnostic process. A non-optimal pupil size may lead to vignetting effects, where image quality decreases towards the periphery. Overcoming these problems often requires careful patient guidance during the examination to ensure optimal fixation. In some cases, additional imaging modalities or repeated scans may be necessary to obtain reliable and meaningful results.

##### Influence of Concomitant Ocular Diseases on RNFL and MRW Imaging 

The impact of ocular comorbidities such as papillary traction, ONH drusen, or papillary edema secondary, e.g., to uveitis on the RNFL and the MRW (Figure 2), is crucial for precise diagnostic imaging. These comorbidities can cause substantial changes in RNFL thickness and MRW, influencing the interpretation of OCT images.

Papillary traction, e.g., due to persistently adherent vitreous limiting membrane can lead to local deformations of the RNFL and/or MRW. These changes can result in an asymmetrical distribution of thickness. Subsequent scans may have an increased likelihood of segmentation errors, as the morphological changes due to traction can impact automated layer segmentation. It is essential to note that automated segmentation by modern OCT devices is highly accurate in most cases. Nevertheless, certain anatomical and pathological conditions occasionally require manual review and adjustment to ensure more precise and reliable results.

Drusen within the optic nerve head also influence both biomarkers. Deposits of extracellular material can cause local thickening and irregularities, posing challenges to segmentation techniques. Papillary edema, a common manifestation of uveitis, results in a global increase in RNFL thickness and MRW especially in the presence of optic disc leakage [9].

Increased attention to potential segmentation errors makes such pitfalls visible and continuous observation of the B-scan in addition to the curve visualization is essential to ensure accurate diagnostic information and proper patient care. 

OCT also provides diagnostically relevant data for other diseases, as these can exhibit pathognomonic findings that enable an accurate diagnosis. The assessment of the peripapillary areas in which the RNFL is diluted is particularly important for this. OCT technology is therefore also used to diagnose underlying neurological diseases such as Alzheimer’s disease, dementia, multiple sclerosis and hereditary optic neuropathies (Figure 3). 

In contrast to glaucoma, RNFL analysis shows a clear temporal preference of RNFL loss in these patients. The observed thinning of the temporal RNFL corresponds with the stronger effects on the papillomacular fibers. Furthermore, OCT can be useful in many other optic neuropathies, e.g., for early detection of damage in the case of optic atrophy of retrobulbar origin [10].

##### The Minimum Rim Width (MRW)

The BMO is a crucial anatomical reference point in the optic nerve head, representing the transition from neuroretinal tissue to the optic nerve. Some special features must be taken into account in the evaluation, especially in comparison to the RNFL.

-
**Differentiation from opticopathies of non-glaucomatous origin:**


The typical glaucomatous thinning of the NRR can be helpful in evaluating BMO-MRW to distinguish it from a simple opticopathy of a different origin. Glaucoma is associated with the development of excavation, leading to a significant decrease in MRW during the course of the disease. Opticopathies of other origins (e.g., AION, optic neuritis) show thinning of the NRR to a much lower extent than in glaucoma, and therefore, BMO-MRW may be less affected than RNFL imaging.

-
**Influence of the optic disc size on the MRW:**


Macro- and micro-optic discs present a particular challenge in ophthalmic examination and challenges in image analysis requiring specialized approaches to accurately interpret OCT data and normative databases parameters may not always account for the variations seen in macro- or micro-discs, emphasizing the importance of considering individualized assessment.

In comparison to microdiscs, macrodiscs show a larger NRR in histological studies [11]. This was also confirmed in BMO-MRW studies using SD-OCT [12]. The micropapillae show a higher density of RNFL in a smaller area, which can lead to the phenomenon of the “crowded disc”. Further, actual disk size varies with race and possibly other demographic characteristics [13]. 

Integrating BMO-MRW into the diagnostic assessment enables a more accurate characterization of optic nerve structure in macro- and micro-optic discs, contributing to improved diagnostic precision. The resulting imaging outcome of BMO-MRW curve can be altered in the case of a macro- and micropapilla (Figure 4). Large normal optic discs can show a flattened amplitude of the curve and the values are in the lower percentile range due to the NRR distribution along a larger scleral canal circumference or the BMO. Conversely, due to the small diameter of the opticoscleral canal, small optic nerve heads usually result in an increase in the amplitude of the curve and BMO-MRW values in the upper percentile range [14].

The literature describes a better diagnostic power for BMO-MRW in macrodiscs to discriminate glaucoma patients from normal controls compared to RNFL and rim area [15]. 

In small optic discs, BMO-MRW and RNFL have similar sensitivity to discriminate patients with glaucoma from normal controls [16]. It is also described that the smaller the BMO is in micro-optic-disc, the more frequently severe visual field damage occurs. This result can be a useful addition for glaucoma assessment [17].

-
**Influence of refraction on MRW**


Myopia is a known risk factor for glaucoma [18]. The diagnosis of glaucoma is often clinically difficult due to anatomical features in the area of the optic nerve head. OCT technology can increase sensitivity and specificity, but requires urgent consideration and knowledge of some special features in image analysis. There is a frequent presence of peripapillary irregularities caused by a loss of the retinal pigment epithelium, the photoreceptor layer and/or the Bruch membrane [19]. Jonas et al. divided them into delta, gamma (peripapillary sclera without Bruch membrane present) and betazones [20]. 

In myopia, the edge of the Bruch membrane can move away from the temporal border of the optic nerve head, so that in a multicenter study, particularly in eyes with a large peripapillary gamma and delta zone with high myopia, approximately 30% of eyes had an undetectable BMO in at least one meridian. In addition, the BMO was most often not visible in areas where glaucomatous neuroretinal loss was most common [21]. 

The proportions were significantly higher in eyes with high myopia than in eyes without high myopia in the glaucoma and healthy control groups (*p* < 0.001). The temporal meridian, followed by the inferotemporal and superotemporal meridian of the optic disc, were the most common sites of an undetectable BMO. Increased axial length (AL), advanced glaucoma, β-parapapillary atrophy and young age were associated with an increased hazard ratio of an undetectable BMO (*p* < 0.032). Therefore, the use of BMO-based parameters in highly myopic eyes may potentially interfere with the diagnostic evaluation of glaucoma.

##### The Retinal Nerve Fiber Layer (RNFL)

Various things must also be taken into account when analyzing the RNFL. 

-
**Influence of refraction on the RNFL:**


Refractive errors of patients have a significant influence on the RNFL. The specific anatomy of hyperopic or myopic eyes therefore often makes imaging and diagnostic assessment with structural comparisons to a normative population difficult.

With increasing AL, a decrease in the thickness of the RNFL is described in all areas except the temporal quadrant. The global RNFL thickness is thinner by 3.086 μm per millimeter AL. The thinning is most pronounced in the superior and inferior regions [22].

Myopic eyes with additional posterior staphylomas may be structurally distorted. The superotemporal and inferotemporal RNFL bundles tend to converge temporally with axial elongation. This can lead to an apparent false-positive thinning in the superior and inferior sectors of the RNFL deviation map when compared to the normative database of non-myopic eyes. It causes more temporally located superior and inferior peaks (or ‘humps’) of the RNFL thickness profile [23]. Myopic OCT findings can therefore be falsely assessed as glaucomatous. Some studies also describe a thicker RNFL in the temporal region in myopic eyes [24], so that a possible thinning in the papillomacular bundle, as can also occur in glaucoma, may be overlooked. In hyperopic eyes (>+3 dpt), a significantly thicker retinal nerve fiber layer was found in Vergleich zu myopen Augen [24]. 

The relationship between hyperopia and RNFL thickness is rarely described in the literature. The higher the degree of hyperopia, the thicker the RNFL. In hyperopic eyes, the RNFL is thicker in the nasal, nasal-inferior, as well as nasal-superior regions compared to myopic eyes. In the global context, there is a close correlation between RNFL thickness and an increase in refractive error [24]. 

Reports have indicated alterations in the configuration of the retinal image in the presence of cylindrical refractive error, where the elliptical shape is observed and varies based on the astigmatism axis [25]. Previous research findings have highlighted a reduction in RNFL thickness specifically in the temporal quadrant, attributed to the increased spatial separation between the fovea and the superotemporal and inferotemporal peaks compared to the emmetrope group. Nevertheless, no significant differences were observed in the global average and non-temporal RNFL thickness [26]. The precise impact of astigmatism on RNFL thickness and optic disc parameters remains ambiguous. 

To reduce these diagnostic problems, some software programs can take better account of the length of the patient’s eyeball by entering the C-scan value (corneal curvature radius) before the first measurement. This will enable even better differentiation between pathological and healthy findings in the future (Figure 5).

-
**Influence of ethnicity:**


Ethnic background is another known influencing factor on OCT biomarkers. Studies have shown that the RNFL is significantly different in some cases between people of different origins [22]. OCT images are evaluated automatically using a normative database, which is, however, subject to restriction to certain ethnicities by the manufacturers. For example, the sensitivity of RNFL and BMO-MRW for the detection of incipient glaucoma was described in the study by El-Nimri et al. as being significantly lower in subjects of African origin than in those of European descent [27]. This emphasizes the importance of considering ethnic differences and that the reference cohort should be representative of multiracial patients.

#### 2.1.3. The Power of OCT Imaging to Detect Glaucoma and Glaucoma Progression

In daily clinical practice, optic nerve head imaging should provide help with two tasks: (1) the ad hoc (with the first exam) classification as either healthy or glaucomatous (or otherwise pathological) and (2) the detection of glaucoma progression over time. For this purpose, it delivers two parameters: The thickness of the RNFL at very high spatial resolution and an overall pattern of distribution of retinal nerve fibers, usually presented as thickness plot. The latter is particularly important for the ad hoc classification of early glaucomatous damage with only a small portion of fibers lost. The distribution of nerve fibers, however, is subject to considerable inter-individual variability. Consequently, an accurate classification between early glaucomatous damage and a non-pathological variant of nerve fiber distribution is difficult to achieve. Features like left/right eye overlay may help, as non-pathological variants tend to be symmetric between eyes, while glaucoma usually introduces asymmetry to the RNFL distribution. However, the fact remains that OCT performs relatively poor in distinguishing early glaucomatous optic discs from healthy discs in a screening setting with only a single examination [28,29].

In contrast, the outstanding strength of OCT imaging, its very high spatial resolution, has the greatest utility when used for longitudinal monitoring of glaucoma progression or conversion from normal to glaucomatous. Then, individual peculiarities of RNFL distribution become, as a static parameter, irrelevant and the micrometer-level resolution enables early detection of structural loss. It has to be considered that due to the high sensitivity of the system also the physiological age-related loss of nerve fibers will be detected. This may be particularly misleading when only global (average RNFL thickness) trend-based regression analysis is performed [30]. 

The performance of OCT for detection of glaucoma progression is particularly strong for early to moderate glaucoma [31]. To fully utilize its technical potential, it is necessary to use progression analysis software (Figure 6). 

In advanced glaucoma, it gradually declines and is eventually lost when a certain maximum thinning of the RNFL or BMO-MRW is reached, the so-called measurement floor. This happens first locally, and then later globally throughout the 360° circumference (Figure 7). Despite the measurement floor being reached, visual function may still be present, resulting in a diagnostic gap, which severely impedes glaucoma progression assessment [32]. Depending on the instrument used, the measurement floor starts below circa 50 µm average RNFL thickness. Alternative structural measures of glaucoma, such as the density of the superficial capillary network of the perifoveal or parapapillary retina reach the measurement floor later (or may have none) and may thus proof more suitable for progression analysis in advanced stage glaucoma (see Section 3.2.1) [32,33,34]. Also the measurement of the ganglion cell layer thickness, often in conjunction with the inner plexiform layer thickness, may be a useful adjunct to RNFL thickness in advanced glaucoma cases [35].

### 2.2. Scanning Laser Ophthalmoscopy 

Scanning laser ophthalmoscopy was the first technology available to quantify the morphology of the optic nerve head and measure neuroretinal tissue thickness. The commercial device was named Heidelberg Retina Tomography (HRT; Heidelberg Engineering, Heidelberg, Germany). It employed laser beam scanning using a 15 × 15° field centered around the ONH. A total of 64 confocal scans were acquired, each with 62.5 µm focus distance from the previous one. From this, a three-dimensional volumetric dataset was reconstructed. The reference plane used for defining the NRR was determined based on a manually drawn contour line of the disc margin. 

The morphometric analysis is, to a great extent, aimed at providing a quantification of what the physician sees on funduscopic examination, such as cup-to-disc ratio and disc area. However, it also measures RNFL thickness—not the true thickness, though, because the lower limit was somehow the artificial reference plane. 

Modern spectral-domain or swept-source OCT devices outperform HRT in both respects, detection of glaucoma in a screening setting as well as in longitudinal analysis of glaucoma progression [36]. Although HRT sales have been discontinued since 2019 and OCT is gradually replacing it, HRT may still be used with confidence for follow up if long-term collected data exist. 

### 2.3. Optic Disc Photographs

Obtaining time series of optic disc photographs has been the gold standard for decades to assess structural glaucoma progression. With the advent of digital image processing, alignment and flicker tools further improved the sensitivity of progression detection. However, disc photographs were later largely replaced by sectional imaging technology, such as SLO and OCT, which allow quantification of neural tissue loss as opposed to a purely qualitative assessment of structural progression. 

Thanks to the robustness of image acquisition, optic disc photographs have nevertheless remained a useful tool in cases where OCT images are not available or difficult to obtain, such as in children or patients with nystagmus. 

In recent years, artificial intelligence has led to a small renaissance of optic disc photographs. AI has proven surprisingly powerful in the classification of healthy vs. pathological optic nerve [37,38]. Thus, in the near future, a simple fundus image may support health care workers in identifying glaucoma suspects or in timing follow-up visits of glaucoma patients.

## 3. Future Directions

The development of ONH imaging technology is heading in several directions. Notably, many of them are not exclusive to the ONH or even to glaucoma but often mirror trends from other fields of ophthalmology or imaging in general. 

They may be grouped into three directions:(1)**The improvement of OCT data quality and quantity,** for example, by increasing resolution, scan speed, reduction in artefacts and layer/landmark segmentation.(2)**The establishment of new biomarkers**, which address current shortcomings of the nerve-fiber-layer-centered ONH assessment, such as specificity and sensitivity at early stages of glaucoma, floor effect at late stages, detection of risk factors for the development/the progression of glaucoma.(3)**The improvement of data analysis,** where artificial intelligence is a highly powerful tool to integrate data of various sources and to take advantage of data ‘hidden’ in images or scan volumes, which goes currently unrecognized by the established analysis algorithms.

### 3.1. Improvement of Data Quality and Quantity

This is currently an evolutionary process that has recently been driven by the introduction of swept-source technology, leading to increased scan speed, and by artificial intelligence (AI)-based algorithms to improve layer segmentation and landmark detection.

○
**Swept-Source (SS) OCT and Enhanced Depth Imaging (EDI) Technique:**


SS OCT represents a more recent advancement in OCT technology, utilizing a tunable laser light source known as the “swept source” to capture in vivo images of ocular structures. This technology offers benefits in comparison to the main benefit of considerably faster scanning speeds when compared to SD-OCT systems. This enhanced speed facilitates the implementation of denser and more intricate scanning patterns.

Using EDI-OCT, it is possible to visualize structures 500–800 μm deeper than with conventional OCT. This characteristic arguably enhances the imaging of deeper ocular structures, such as the LC and choroidea. Presently, there is a lack of adequate data regarding the diagnostic efficacy of LC imaging using SS-EDI-OCT, making it premature to endorse its clinical application.

○
**Adaptive optics (AO):**


A better visualization of RGCs and the LC is of great interest for pathophysiological studies of glaucoma. The visualization capabilities are limited by optical aberrations caused by the eye’s optics, which restrict the lateral resolution of these systems. AO technology promises to expand the capabilities of existing imaging modalities. The impact of wavefront aberrations on optical systems like OCT manifests in focus distortion, potentially reducing image resolution and contrast. This influence is observed not only in image properties but also in the coupling of light into optical fibers, affecting resolution and signal-to-noise ratio in a fiber-based OCT system, for example. Particularly in the in vivo environment, aberrations can vary from sample to sample, necessitating dynamic correction. AO technology provides a solution by enabling the correction of aberrations, thereby restoring the optimal performance of the optical system. This allows the capture of detailed three-dimensional images of retinal cells/structures at the microscopic level.

The initial application of AO in ophthalmic practice was documented in 1997 through merging it with a fundus camera [39]. Liang et al. pioneered the integration of a Shack-Hartmann wavefront sensor with a deformable mirror for aberration correction, leading to the acquisition of high-quality retinal images. Until now, AO has been integrated with flood illumination ophthalmoscopy, conventional fundus cameras, scanning laser ophthalmoscopy (AO-SLO), as well as OCT (AO-OCT).

There are currently only a few studies that have examined glaucoma eyes using the AO technique. These few studies have mainly examined the macular area, so that there are hardly any publications on examination results in the area of the optic nerve head.

○
**AO and RNFL imaging:**
-Choudhari et al. compared the image characteristics of the RNFL between glaucoma patients and healthy controls using AO-SLO. Imaging was performed one degree away from the optic disc margin in glaucoma patients and healthy controls. The field of view was 1.3°. A comparison of early glaucoma with healthy subjects showed a significant difference in reflectivity (0.51 and 0.56, respectively, *p* < 0.001), but not in the structure of the RNFL bundles. In patients with moderate and severe glaucoma, no RNFL bundles were recognizable and the RNFL reflectivity did not differ from that of patients with early glaucoma (*p* < 0.11). The authors therefore conclude that the enhancement (%) of RNFL images recorded with AOSLO could be an objective indicator of early glaucoma [40].
○
**AO and LC imaging:**
-AO can support studies on the biomechanics of the LC.-Confocal AO-SLO imaging enabled high-resolution 2D visualization of the LC in healthy eyes and quantitative analysis of pore geometry with small intersession and larger intersession variability. However, the authors conclude that AO-SLO imaging could be used to measure and track changes in laminar pores in vivo during glaucomatous progression [41]. In glaucoma, LC pores seem to be elongated measured with flood-illumination AO fundus (FIAO) camera [42].-Nadler et al. characterized the in vivo three-dimensional LC microarchitecture of healthy eyes using AO spectral-domain OCT (AO-SDOCT).-The results showed regional differences in the microarchitecture of the LC by quadrant and in depth, which should be considered in further studies of the LC. Nevertheless, they reported poor repeatability in measuring lamina volume.-According to the authors, this is due to the narrow depth of focus, which hindered the detection of the scleral canal opening, which is a standard feature for delineating the optic disc [43].
○
**AO and retina vessel imaging:**
-AO can measure various parameters in retinal blood vessel imaging, including vessel diameter (inner and outer), vessel wall thickness, vessel distribution characteristics (vascular branching) and blood flow parameters. To our knowledge, there are currently no publications on peripapillary vascular changes in glaucoma measured by AO.-Integrating AO into clinical settings poses challenges. AO-OCT has a limited small field of view, to gain a larger overview one would need to create a compound image, which can be time consuming. Moreover, because of the restricted depth of field, it might be required to combine volumes or modify the focal plane to achieve comprehensive visibility of retinal structures at various depths. This proved to be technically challenging, especially in the region of the LC for delineating the boundaries [43].-Nevertheless, AO undeniably stands out as an outstanding research tool, providing the potential to visualize the retina at cellular level. Incorporating AO into clinical settings presents challenges.


### 3.2. Novel Biomarkers

#### 3.2.1. Optical Coherence Tomography Angiography (OCT-A)

Impaired blood supply is considered a potential pathogenic factor in the development of glaucoma. Reduced blood flow to the optic nerve head can cause and/or worsen the degeneration process of the RGC and the optic nerve. 

The blood supply to the retina and optic nerve is primarily derived from the central retinal artery (CRA). Emerging from the ONH, the CRA divides into four main branches, which then give rise to smaller arterioles. These smaller vessels form a complex three-layered capillary network comprising superficial, intermediate, and deep vascular plexuses. An additional plexus, called the radial peripapillary capillary plexus (RPC), exists. The RPC plexus is the most anterior layer of capillaries within the thickest region of the RNFL.

Recognizing circulatory disorders could be crucial for the early detection and treatment of glaucoma.

The role of blood flow measurement in glaucoma has been investigated and described even before the advent of OCT-A, using various other techniques. The works of Harris et al. have significantly contributed to understanding the importance of ocular blood flow in the pathogenesis and progression of glaucoma. They have demonstrated that patients with glaucoma often exhibit reduced ocular blood flow, particularly in the peripapillary retina and choroid [44,45]. These findings suggest that impairment of the microcirculation may lead to ischemia and ultimately result in retinal ganglion cell loss. Furthermore, Harris et al. have investigated the use of various imaging techniques to assess ocular blood flow in glaucoma, including color Doppler imaging (CDI) and laser speckle flowgraphy (LSFG) [46]. 

Advancements in imaging techniques such as OCT-A and other advanced technologies allow for a more precise depiction of the optic nerve’s blood supply. 

OCT-A offers non-invasive, detailed insights into the microvascular structures of each vascular network by identifying changes in reflectivity resulting from the movement of blood cells within vessels. It provides a means to visualize and quantitatively assess retinal vessels with excellent reproducibility and repeatability [47] including the determination of perfusion density (PD) and vessel density (VD). Segmentation capabilities of OCT-A enable the performance of separate analyses for peripapillary retina and ONH regions, as well as the deeper vessels of the choroid. Quantifications at the segmentation level of the radial peripapillary capillary network are most frequently found in the literature. The RPCs are located within the RNFL up to 4–5 mm from the ONH. 

OCT-A has detected a significantly reduced vessel density (VD) in the papillary and peripapillary region in cases of glaucoma compared to healthy individuals with good discriminatory power to differentiate normal from glaucoma eyes (Figure 8). The VD decreases with increasing severity of glaucoma. VD appears to have higher variability [48] and lower sensitivity compared to peripapillary RNFL. The determination of VD has excellent and good reproducibility in most retinal layers, both in healthy subjects and in glaucoma patients, regardless of the severity of the disease [49].

Furthermore, associations between vascular parameters and structural (RNFL) [50] and functional parameters (perimetry) [51] of glaucoma severity have been demonstrated. 

Although OCTA has already been shown to be useful in glaucoma, it has not yet been able to find its way into standardized use in everyday clinical practice as it is not glaucoma-specific. It has also shown the ability to detect vascular changes in patients with systemic diseases such as Alzheimer’s disease [52], systemic hypertension [53] and multiple sclerosis [54]. Overall, these pathologies lead to a lower retinal and choroidal vascular density. Furthermore, OCTA appears to encounter the “floor effect” at a more advanced stage compared to OCT [55]. This suggests potential benefits for handling advanced glaucoma cases, particularly even when visual fields tend to be less reliable. OCT-A could provide valuable prognostic information in these cases and support treatment decisions.

The significance in the diagnosis of various ophthalmological clinical pictures, the comparability of the analyses between the different device manufacturers and the standardized procedure for the evaluation and quality control of OCT-A data are important aspects that are still under discussion. Furthermore, there are no normative datasets widely accessible to provide clinicians with effective information regarding the extent of damage in their patients. Future research may focus on integrating this technology to understand not only structural but also functional aspects of vascular supply.

In view of the improved visualization possibilities of the vascular structures in the deeper retinal layers, particular interest has been shown in examining the deep layers with the short posterior ciliary artery, which also perfuses the deep structures of the optic nerve head (ONH). SS-OCT-A enables enhanced visualization of the deeper vascular layers, particularly the choroid. This could contribute to a better understanding of the role of choroidal perfusion in the development and progression of glaucoma and potentially identify prognostic markers. There are several reports that reduced choroidal vascularization may be associated with the development of glaucoma. At the level of the choroidal vasculature ***microvascular dropouts*** within the beta-zone para-papillary atrophy have been described (Figure 9) [56]. They are related to a higher risk for glaucoma progression and show a spatial and temporal relation to disc hemorrhages. Moreover, they are more likely to occur in highly myopic eyes [57].

A multicenter prospective cohort study was conducted to explore the role of vessel parameters in identifying individuals at risk of developing normal tension glaucoma (NTG) among NTG suspects. Patients with suspected normal tension glaucoma, exhibiting either baseline microvasculature dropout (MvD) or reduced laminar deep vessel density (VD) on OCT-A, demonstrated an increased risk of conversion [58]. Identifying such high-risk patients could enable early treatment and closer monitoring. Longitudinal studies are essential to uncover the association between parapapillary deep-layer microvasculature dropout and systemic as well as ocular factors.

The integration of OCT-A data with other imaging and functional tests could enhance the prognostic value. Future research should focus on developing multimodal approaches that consider structural and functional aspects of vascular supply to optimize the prognosis and management of glaucoma patients.

However, the development of integrated normative datasets that combine OCT-A parameters with other imaging biomarkers and functional measures is a complex task that requires large, well-characterized patient cohorts and advanced statistical modeling techniques.

In conclusion, OCT-A, particularly with the SS technique, has the potential to serve as a valuable tool for the prognosis of glaucoma. However, further research is needed to validate the prognostic value of OCT-A parameters, develop standardized evaluation protocols, and optimize integration with other diagnostic modalities. The establishment of robust normative datasets for OCT-A parameters, especially for the deeper vascular layers, is a critical step towards the clinical application of this technology for the diagnosis and management of glaucoma. Addressing the challenges in establishing these datasets requires collaborative efforts among researchers, device manufacturers, and professional organizations to ensure the reliability and generalizability of OCT-A data.

#### 3.2.2. Polarization-Sensitive (PS)-OCT

Conventional OCT provides information about tissue by measuring the intensity of backscattered (=reflected) light. However, much more than just backscattering can happen to light when it travels through tissue. One of these things that go undetected by conventional OCT is a change in the oscillation plane of the light wave. Some tissues in the eye have the ability to change the oscillation plane and sometimes they also variably change the speed and direction of the light beam depending on the oscillation plane of the incoming light wave. This phenomenon is called birefringence and characteristic for, among others, the retinal nerve fiber layer, cornea and sclera. It is, at least in biological tissue, the result of a certain tissue composition where regularly arranged structures such as axons are embedded in a matrix with a different refractive index. In fact, experimental data from animal experiments show that the longitudinal cytoskeleton (mainly microtubuli) inside the axon is responsible for the birefringent properties of the RNFL [59]. 

The predecessor of PS-OCT was scanning laser polarimetry (SLP; commercial name GDx (Carl Zeiss Meditec, Dublin, CA, USA)). It was, like HRT, based on a scanning laser ophthalmoscope but measured the degree of phase retardation of polarized light in the RNFL. Regarding the ability to detect glaucoma in longitudinal observation it was outperformed by SD-OCT [60] and the increasingly widespread use of OCT finally ended the era of SLP. However, the quantification of birefringence remains an interesting biomarker, particularly in very early glaucoma, because a disruption of axonal microtubule and a subtle loss of ganglion cell axons may already produce a reduction in birefringence before a detectable reduction in RNFL thickness occurs [59,61].

PS-OCT is a technology that can be added to both, spectral-domain or swept-source technology. Thus, it keeps up with the technical development of OCT in terms of scan speed, resolution and field of view and it provides (other than SLP) polarization-related parameters in 3D and in parallel to the reflectivity data of conventional OCT (Figure 10). 

A recent study from Steiner et al. [62] demonstrated a diagnostic accuracy of the PS-OCT-derived parameter phase retardance similar to RNFL thickness in early stage glaucoma patients (visual field mean defect (MD) < 6 dB) compared to age matched controls. A superiority, however, in the detection of very early defects has so far only be shown in animal experiments using non-human primates [63,64]. 

It should be noted that PS-OCT shares the same limitation with conventional OCT and OCT-A -derived parameters in that alterations from the physiological pattern are not specific for glaucoma. Diabetic retinopathy has recently been shown to produce a significant reduction in phase retardance and birefringence in the peripapillary RNFL [65].

In addition to RNFL, strong birefringence has been observed in the LC and the sclera [66,67]. As this phenomenon is related to the arrangement of collagen fibers, it is conceivable that aberrant birefringence properties may indicate aberrant mechanical properties of the LC and the peripapillary sclera. These in turn are factors that very likely influence the translation of elevated IOP and IOP fluctuation amplitude into mechanical stress to axon bundles (see also Section 3.2.5).

#### 3.2.3. Measurement of RNFL Reflectance with Visible-Light (VL)-OCT

In contrast to the conventional RNFL thickness measurement, the assessment of RNFL reflectivity (the amplitude of the OCT signal) may be a biomarker, which is in part influenced by the ultrastructural properties of the RNFL axon bundles. Experimental data using spectroscopy in a rat model of glaucoma with longitudinal follow-up describe changes in RNFL reflectivity prior to the onset of RNFL thinning [61]. The amplitude of the reflectance change may depend on the wavelength of light used [68]. This opens up a role for visible-light OCT, which employs a light source emitting visible light as opposed to the near-infrared light of standard OCT devices. The shorter wavelength of the VL-OCT allows slightly higher resolution and allows to gain additional contrast from utilizing the ratio between visible and near-infrared light OCT signal. At the same time, shorter penetration depth, higher scattering in opaque optical media (dry eye, cataract) and a more challenging image acquisition due to reduced patient compliance when the eye is exposed to a visible light scan beam are significant obstacles in the way of a routine use in glaucoma clinics. Song et al. [69] published a recent pilot study using a dual channel (visible (green to yellow) and near infrared light) OCT. As a proof of concept, they showed in a cross-sectional study good discrimination between controls and glaucoma patients, with slightly better performance for the glaucoma suspect/preperimetric glaucoma group compared to standard OCT RNFL-thickness measurement. The hypothesized higher diagnostic accuracy needs to be proven in longitudinal studies.

Another interesting feature of using visible light is the ability to measure oxygen levels in the optic nerve head or peripapillary blood vessels [70]. 

#### 3.2.4. Retinal Nerve Fiber Layer Optical Texture Analysis (ROTA) 

There is increasing evidence that glaucoma disease also leads to defects in the papillomacular bundle, even in the early stages of glaucoma [71,72]. As the loss of the papillomacular bundle has a direct impact on central vision and leads to a reduced quality of life, it is important to use new tools such as optical ROTA for early detection of region-specific defects. ROTA refers to a method that takes a new approach by analyzing the texture and microscopic structures and patterns within the RNFL rather than focusing on its thickness, particularly within the papillomacular bundle with excellent accuracy, reproducibility, and sensitivity [73]. This approach may provide valuable insights into the early stages of glaucoma and potentially improve the ability to detect subtle changes in the texture of the retinal nerve fiber layer associated with the disease. However, as this is a new development, real-world data on this technology is currently lacking.

#### 3.2.5. Imaging of the LC and Peripapillary Sclera

The LC is considered the primary site where elevated IOP or IOP-fluctuation translates into mechanical stress to the retinal ganglion cell axon and the surrounding glia. The architecture and biomechanics of the LC and the peripapillary scleral tissue are the main factors that determine this translation. This applies to both the gross morphological characteristics of the LC, such as position, size, shape and pore morphology as well as the ultrastructural collagen fiber composition of the LC and the peripapillary scleral tissue. 

LC morphology is subject to considerable change during progression of glaucoma [74]. Recent progress in OCT technology, including SS-OCT and enhanced depth imaging, significantly improved the assessment of LC morphological characteristics, allowing the reliable measurement of parameters like LC depth, LC curvature and depth of anterior LC insertion [74]. 

One rationale for the clinician behind LC imaging could be the identification of morphological risk factors for glaucoma progression or conversion from ocular hypertension to glaucoma. One prospective study from Ha et al. [75] in fact demonstrated that a greater posterior bowing of LC at baseline examination in early POAG patients was associated with a significantly faster visual field deterioration during the subsequent 3.5-year follow-up. Structural alterations in the LC may be of particular importance for normal tension glaucoma (NTG) patients. Among patients with focal LC defects, the prevalence of NTG was 33% as opposed to 9% in the group without LC defects [76]. And among Asian NTG patients with initial parafoveal visual field defects LC alterations such as LC thinning, LC defects and shape distortion were significantly more common than in NTG patients with more peripheral visual field defects [77].

Another advantage of LC imaging could be the detection of structural alterations indicative of glaucoma in cases where glaucoma detection is otherwise difficult, such as in high myopia. The presence of LC defects and choroidal microvascular dropouts (see Section 3.2.1) are two such biomarkers, which were more frequently observed in patients with high myopia and glaucoma [57,78].

The application of PS-OCT (see Section 3.2.2) to LC imaging adds valuable information on the collagen fiber architecture of the LC and the peripapillary sclera. In conjunction with the morphology data from conventional OCT this may open the possibility to deduce the biomechanical properties of these tissues from the imaging data [66,67].

### 3.3. Improvement of Data Analysis—The Role of Artificial Intelligence (AI)

The utilization of various techniques and biomarkers in glaucoma diagnostics has resulted in a growing influx of data, forming the foundation for AI-driven analysis methods. AI algorithms are characterized by their ability to search through extensive datasets and enable the integration of various data combinations. This capability has paved the way for the introduction of AI in glaucoma diagnostics, which can be traced back to early efforts that primarily focused on the utilization of fundus photographs. These initial algorithms were designed to extract features, such as cup-to-disc ratio, RNFL, and other indicators of glaucoma risk, offering a more rapid and objective assessment of images compared to traditional manual methods.

The emergence of deep learning techniques represented a significant advancement in glaucoma diagnostics. Convolutional neural networks (CNNs), in particular, demonstrated remarkable ability in discerning intricate patterns within fundus photographs. Researchers achieved outstanding results in identifying specific glaucoma features by training these networks on extensive datasets, thereby enhancing the accuracy of diagnostic processes.

Incorporating AI into comprehensive glaucoma diagnosis has the potential to revolutionize the monitoring of cellular-level changes, such as the loss of RGCs in the RNFL. Advanced AI algorithms, especially deep learning models, possess the capability to detect subtle alterations in the RNFL that may serve as early indicators of glaucomatous damage. By exposing these models to vast datasets comprising OCT and OCT-A images, along with relevant clinical data, AI can be trained to recognize patterns and features associated with RGC loss. This breakthrough could facilitate earlier detection of glaucoma and enable more precise tracking of disease progression at a cellular level.

The combination of perimetry results, OCT images, and eventually OCT-A datasets has facilitated the development of methods for predicting potential disease progression. The incorporation of OCT volume scans ensures greater data consistency, covering a broader tissue area or volume, which is pivotal for the dependability of AI algorithms. The three-dimensional insights from volume scans provide a more comprehensive foundation for AI algorithms to identify patterns and comprehend intricate relationships among distinct retinal layers or tissue structures. AI contributes to refining segmentation detection accuracy, thereby mitigating variability arising from inaccurate segmentation.

Despite promising strides, challenges persist in the realm of AI in glaucoma diagnostics, encompassing issues such as data privacy, standardization, and the imperative for meticulous algorithm validation. Deep learning models, such as CNNs, are often considered “black boxes” due to their complex architecture and decision-making processes, which can hinder the trust and acceptance of AI-based solutions by clinicians and patients. Developing explainable AI models that provide insights into the reasoning behind their predictions is an ongoing challenge. Rigorous validation of AI algorithms through large-scale clinical trials is necessary to establish their clinical utility and safety. Integrating AI algorithms into existing clinical workflows requires the development of user-friendly interfaces, seamless integration with electronic health record systems, and the training of health care professionals to interpret and act upon AI-based recommendations.

Nevertheless, the integration of artificial intelligence into glaucoma diagnostics signifies a noteworthy advancement towards more accurate, efficient, and personalized approaches in ophthalmology. The ongoing assimilation of AI algorithms into clinical practices undoubtedly stands to optimize early glaucoma detection and enhance treatment modalities, safeguarding the vision of patients. Collaborative efforts among researchers, clinicians, industry partners, and regulatory bodies are essential to realize the full potential of AI in advancing glaucoma care. By harnessing the power of AI, clinicians can make more informed decisions, leading to improved patient outcomes and a better understanding of the complex nature of glaucoma.

## 4. Summary

This review sheds light on various imaging-based diagnostic approaches on the optic nerve head that go beyond conventional examination techniques such as fundoscopy. A central focus is on state-of-the-art technologies and advancements. From OCT to OCT-A and the integration of AI in glaucoma diagnostics, diverse techniques are explored to enhance precision and efficiency in diagnosis.

When evaluating glaucoma imaging, the utmost care and attention are required. It is crucial to adopt a thorough and thoughtful approach to identify potential pitfalls, especially in light of individual patient peculiarities. Only through this careful, personalized approach can potential challenges in imaging be recognized and appropriately integrated into the evaluation.

Individual differences in eye anatomy and other patient-specific factors can impact image quality and lead to interpretative difficulties. Therefore, it is of paramount importance to be vigilant for possible disturbances or irregularities, for example, in B-scans with the performed segmentation, during the evaluation to ensure a standardized, high-quality, and valid result interpretation.

This not only enables a reliable diagnosis but also contributes to precisely monitoring changes in the course of the disease.

Looking ahead to the future, further exciting and promising developments are on the horizon. New techniques such as AO, PS-OCT, and innovative biomarkers, e.g., vessel density are expected to significantly expand our diagnostic spectrum. Through these advancements, we anticipate not only more advanced diagnostic capabilities but also a more comprehensive foundation for a better understanding of the pathogenesis of glaucoma.

The integration of different diagnostic modalities, such as perimetry, fundus photography, OCT-volumetric scans supported by AI technology, not only improves diagnostic accuracy but also facilitates the development of precise predictive models for disease progression. Despite ongoing challenges, including addressing privacy concerns in the application of AI in diagnostics and the need for standardization and careful validation of algorithmic approaches, the integration of AI marks a significant step toward more precise, efficient, and personalized diagnostic approaches in ophthalmology, encompassing a broader spectrum of diagnostic tools.

The diverse spectrum of diagnostic possibilities and the use of innovative technologies contribute to optimizing patient care and quality of life, hopefully alleviating socio-economic consequences.

## Figures and Tables

**Figure 1 jcm-13-01966-f001:**
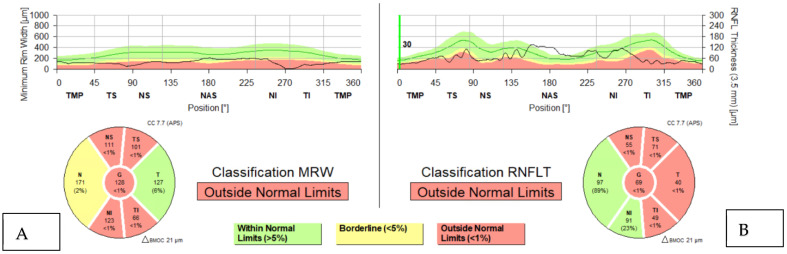
OCT findings typical of glaucoma: (**A**) Amplitude reduction in the MRW increased temporal inferior. (**B**) Glaucoma-typical thinning of the retinal nerve fiber layer RNFL temporal inferior > temporal superior.

**Figure 2 jcm-13-01966-f002:**
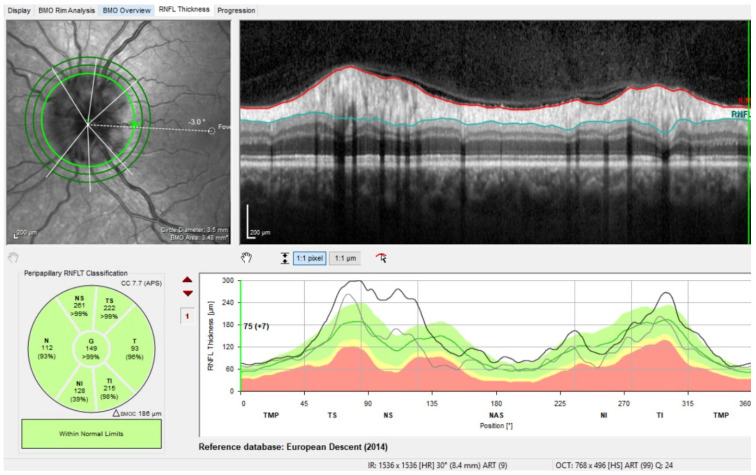
OCT-RNFL progression: Increasing thickness of the RNFL due to new papilledema in posterior uveitis.

**Figure 3 jcm-13-01966-f003:**
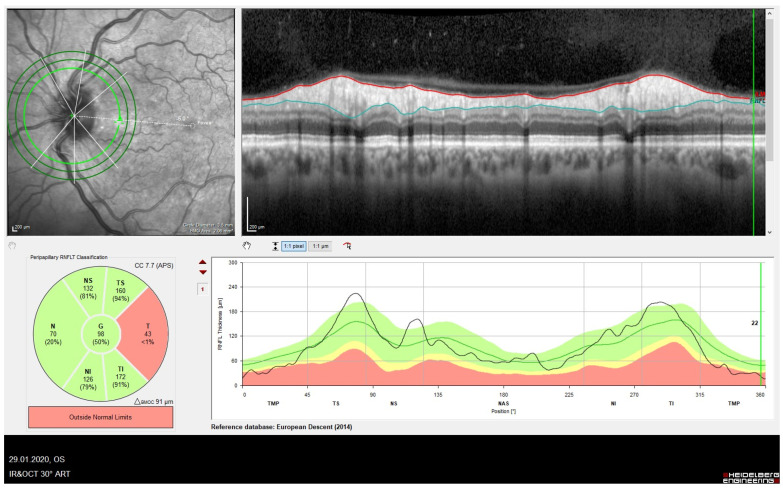
OCT-RNFL: Emphasized temporal thinning of the RNFL in a hereditary opticopathy (LOHN—Leber’s hereditary optic neuropathy).

**Figure 4 jcm-13-01966-f004:**
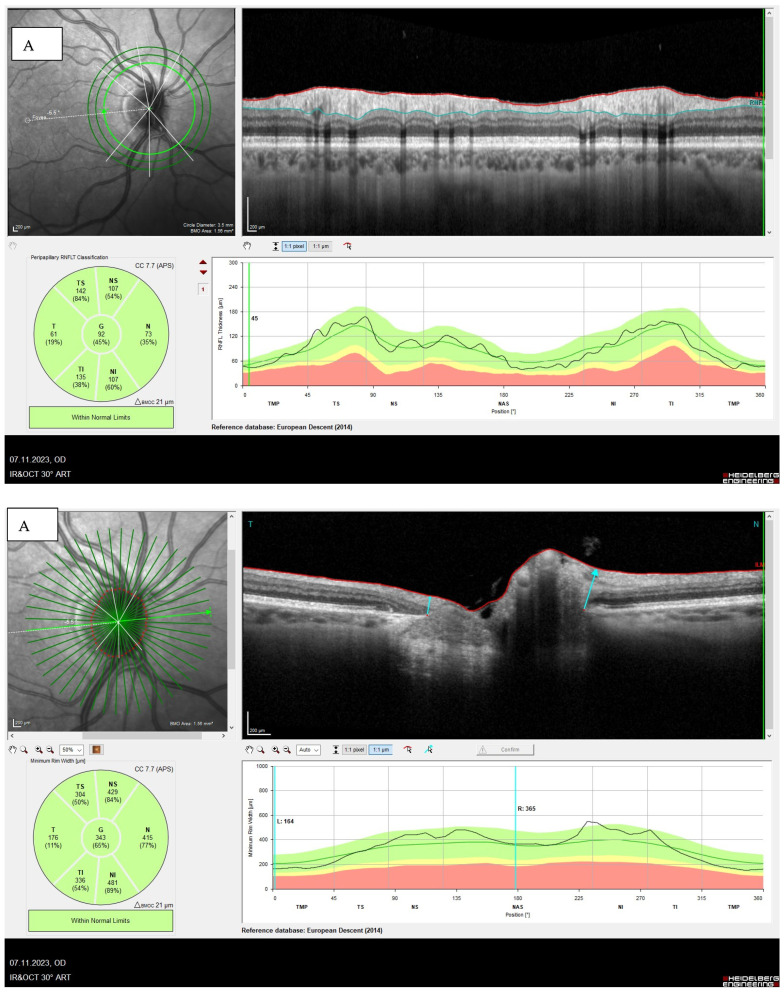
(**A**) Micropapilla (BMO area 1.56 mm^2^) with typical increased amplitude reduction in the MRW (inferior) due to the small diameter of the optic scleral canal. This results in values in the upper percentile range. (**B**) Macropapilla (BMO area 2.28 mm^2^) with MRW values in the lower percentile range due to the distribution of the NRR in a large scleral canal (inferior). The RNFL appears age appropriate (superior).

**Figure 5 jcm-13-01966-f005:**
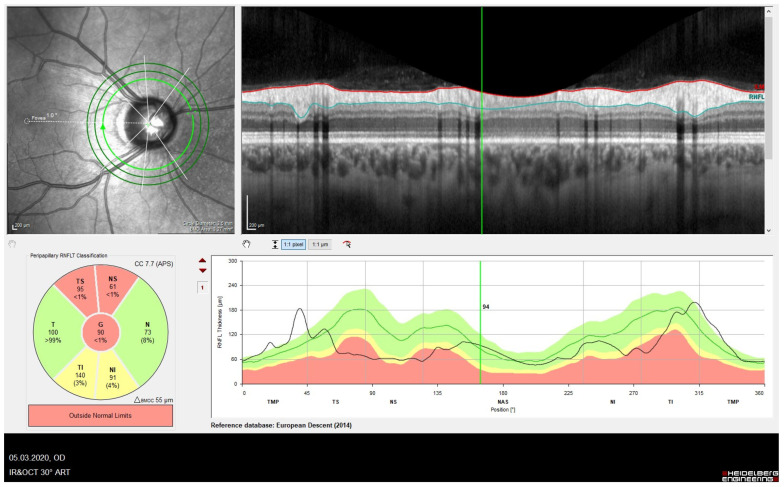
Temporal displacement of the RNFL peak. This results in a finding below the age norm in the temporal inferior and temporal superior regions. In this example, a corneal curvature default of 7.7 mm was preset. The patient had a flatter curvature (8.0 mm). This leads to the displacement of the peaks with false-positive glaucoma findings.

**Figure 6 jcm-13-01966-f006:**
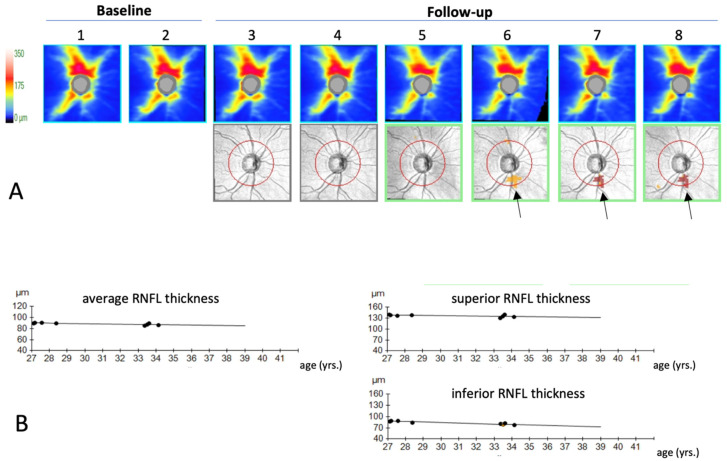
Progression analysis software enables the detection of subtle RNFL thinning over time (this example shows the Guided Progression Analysis software (version 7.0.1.286; Carl Zeiss Meditec, Dublin, CA, USA)). (**A**) An event-based analysis of peripapillary RNFL maps. All images (top row) are registered and differences in RNFL thickness relative to baseline are indicated in the deviation map (bottom row, black arrows). A first occurrence of a significant difference is indicated in yellow, if the deviation is reproduced in the next images the deviating superpixels are shown in red. Event-based analyses indicate deviation from baseline (or a normal reference database) with a relatively small number of samples. They, however, do not provide a measure of the progression rate over time. (**B**) Trend-based progression analysis uses a linear regression analysis to provides a slope for the regression line, which is given in micrometer loss per year format and it provides a *p*-value for the probability that the slope of the regression line is different from zero (no change over time). In contrast to the event-based analysis, more data are required to detect a significant change and spatial resolution is lower than in (**A**).

**Figure 7 jcm-13-01966-f007:**
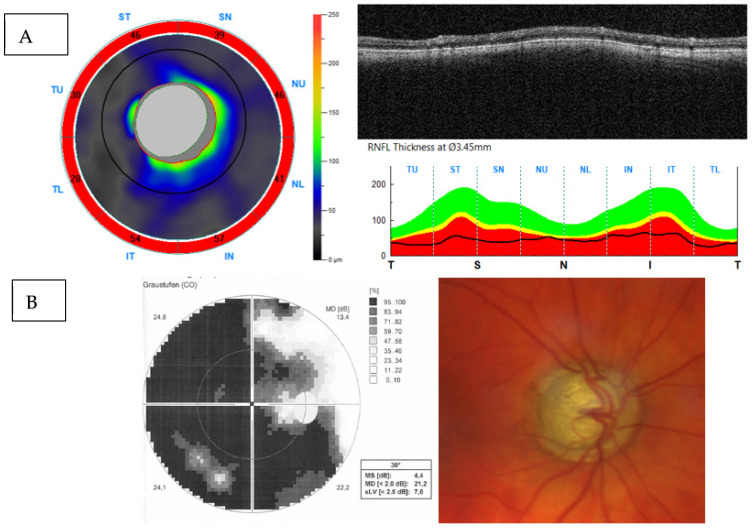
Very advanced glaucomatous atrophy of the optic nerve head with the measurement floor reached in peripapillary RNFL-thickness scan (**A**). Note that the 30° visual field still shows light perception in the temporal field (**B**).

**Figure 8 jcm-13-01966-f008:**
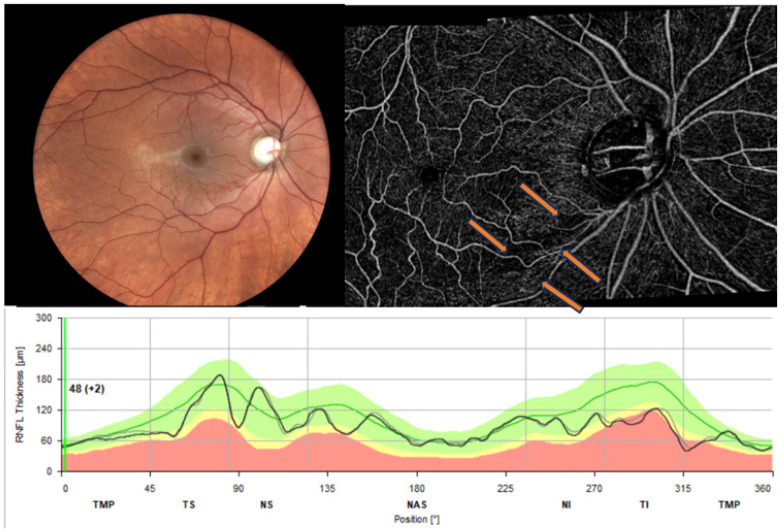
Typical glaucomatous finding of reduced vessel density (arrows) in the temporal inferior area, corresponding to the RNFL loss temporal inferior.

**Figure 9 jcm-13-01966-f009:**
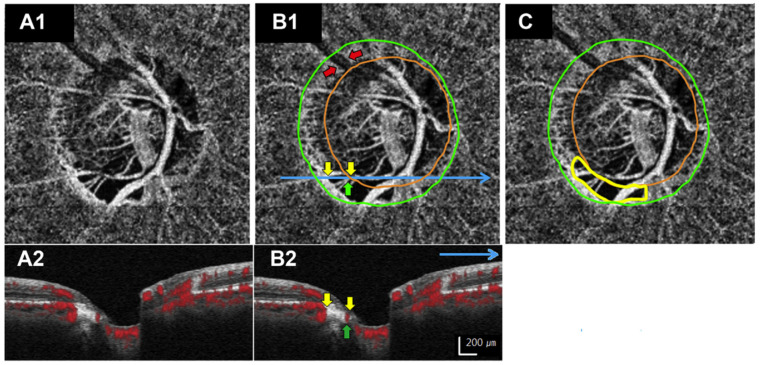
(From Suh et al., Ophthalmology 2016 [56]): Microvascular drop out of the choroid within a beta-zone parapapillary atrophy. All images are from the same optic nerve head. (**A1**,**B1**,**C**) show the same enface choroidal OCT-A slab. The blue line in (**B1**) indicates the position of the OCT-A B-scan shown in (**B2**) (the same scan in shown in (**A2**) without annotations). The colored encircled areas in (**B1**,**C**) indicate the position of the optic nerve head margin in orange, the outer boundary of the beta-zone parapapillary atrophy in green (both were determined on the corresponding SLO image) and the microvascular dropout in yellow. The yellow arrows in (**B1**,**B2**) point to the level of the pigment epithelium, where the beta-zone atrophy was detected; the green arrow points to a major vessel and the red arrows to a shadowing artefact, which were both excluded from the microvascular dropout detection.

**Figure 10 jcm-13-01966-f010:**
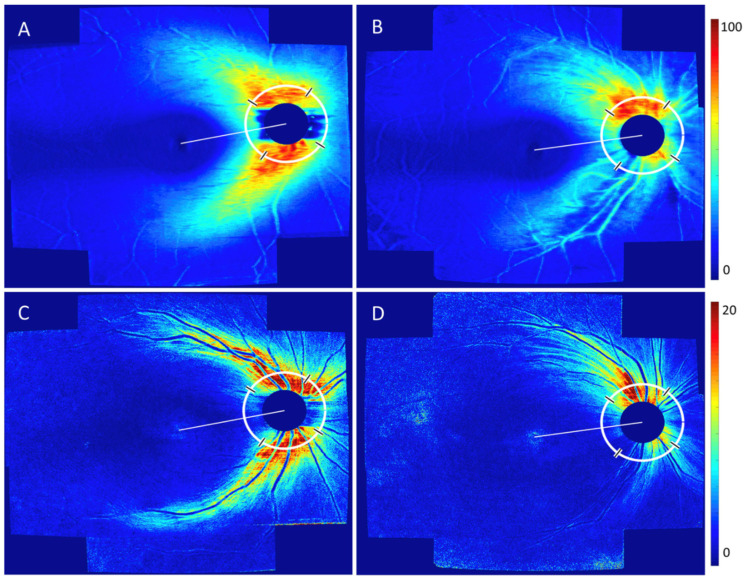
(From Steiner et al. IOVS 2022 [62]) Comparison of conventional OCT (**A**,**B**) and polarization-sensitive OCT (**C**,**D**; phase retardance) of the peripapillary RNFL between a glaucoma patient (**B**,**D**) and an age-matched control patient (**A**,**C**). Both modalities show an infero-temporal RNFL defect. Note the much higher resolution of individual nerve fiber bundles in the PS-OCT images.

## Data Availability

Dataset available on request from the authors.

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
