# Peer review of "Current Status and Future Perspectives of Optic Nerve Imaging in Glaucoma"

_jcm, 2024, doi:10.3390/jcm13071966_

Round 1

Reviewer 1 Report

Comments and Suggestions for Authors

This review has presented the currently available imaging modalities for optic nerve imaging in the diagnosis of glaucoma. The potential of OCT-A over conventional OCT and other alternative imaging techniques has also been accentuated. In addition, the information provided in this review highlights the capabilities of artificial intelligence (AI) in diagnosing glaucoma with greater accuracy and efficiency. However, the authors are requested to address some suggestions that may help enhance the quality of this review.

 Major:

1.     Do authors think that incorporating AI into extensive glaucoma diagnosis can help monitor cellular level changes, for instance, the loss of retinal ganglion cells in the retinal nerve fiber layer (RNFL)? Please discuss this in section 2.3.

2.     Please discuss the prognostic potential of OCT-A in glaucoma treatment.

Minor:

1.     Please define any abbreviation of its first mention and use only the abbreviations later in the text.

2.     Incorporate figure numbers in the text to quote relevant information.

3.     Please label each individual figure with letters (e.g. A, B, C) instead of using left and right mentions, and change the figure legends accordingly.

4.     Figure 4 is not as clear as other figures. Please replace it with a better-resolution figure.

5.     The citation on page 20, line 703 needs to be modified as other citations.

6.     The sentence on page 20, line 714 requires to be completed.

Comments on the Quality of English Language

Moderate editing of English language is required.

Author Response

Dear Reviewer,

We would like to express our sincere gratitude for your thorough review and the constructive feedback you provided for our manuscript "Current status and future perspectives of optic nerve imaging." Your valuable suggestions and insightful comments have been instrumental in enhancing the quality of our work.

We have carefully considered each of your recommendations and have critically revised the manuscript accordingly. We believe that the incorporation of your suggestions has significantly improved the clarity, coherence, and overall impact of the paper. Your input has helped us to better highlight the key points and present the information in a more accessible manner for the readers.

To facilitate your review of the changes made based on your suggestions, we have uploaded a revised version of the manuscript with the modifications clearly highlighted. This version incorporates the changes made in response to the feedback received from both reviewers. We hope that this will simplify the process of evaluating the revisions and ensure that all the necessary improvements have been addressed.

We are confident that the revised manuscript now offers a more comprehensive and valuable resource for those interested in the current state and future directions of optic nerve imaging. The changes we have made based on your feedback will undoubtedly benefit the readers by providing them with a clearer understanding of the subject matter and its implications.

 If you have any further suggestions or require additional changes, please do not hesitate to let us know. 

Sincerly,
Claudia Lommatzsch
On behalf of all co-authors

Reviewer 2 Report

Comments and Suggestions for Authors

I would like to state that the article was so good , informative , educative and contained all relevant information and  I thought that all we had to do was to answer your single word questions.However now that you want further information , I am adding point wise my comments on the article. 

This is a review article and has consistently explained and addressed the title of the paper:Current status and future perspective of Optic nerve imaging in glaucoma".The authors have not addressed any specific question in the paper . The article does not contain any original work of the authors but has  dealt with each imaging modality used currently  and what is expected to come in glaucoma and they have given the latest information in existing literature pertaining to that imaging device in a comprehensive  way ,adding information about robustness of each equipment,The authors have pointed out what  more is still required to help diagnosis of glaucoma, one of which deals with glaucoma and myopia. In this context, one information which needs to be added is limited  normative data  of OCT in children . Work has been done in this field by only one author in India and has been incorporated in only one OCT machine, NIDEK, which should be added as this is the only data available in the world about OCT in children. Another area which has not been covered fully is the role of measuring blood flow in glaucoma- this has been dealt with in  OCT angiography only. Some excellent work done by Alan Harris should be added with its clinical utility.Since there are many firms supplying OCTs , the authors could explain the differences in the print outs of the more commonly used ones and all extra information given by some machines which is more helpful to the readers and ophthalmologists.  As stated earlier, the authors have serially dealt with all information in existing literature , explaining its pros and cons and how robust is that information for diagnosing glaucoma in todays' day  and age.They have added all the newer modalities like Adaptive Optics and how it can give information about the Lamina cribrosa and retinal ganglion   cells.This information is easily not found and has added to the relevance of the paper addressing the need to incorporate the latest technological advances in diagnosing  early glaucoma , advanced glaucoma and also glaucoma progression. Since this paper is not a comparison of modalities, the question of controls does not arise. The question raised by the editor in Q5 is not totally relevant to this article which has collated data - the old and new and information which is in the pipeline ,which has still  to be added to textbooks and a lot is in experimental stage.The conclusions about the relevance, clinical utility, robustness , applicability of each equipment in diagnosing the two difficult areas of glaucoma- the early and advanced and diagnosing progression has been adequately addressed. Infact , a reader interested in this subject will get good relevant information about the same. The references are adequate. Only one OCT  picture has been shown - The HEIDELBERG. The pictures are of good quality and appropriate to the text but if similar pictures of other commonly used machines are also added to show the differences amongst them and their advantages , it will help the reader and will enhance the educative value of the article. I hope the review has  now been completed .It would be advisable to ask  the reviewers to address these questions  at the very first instance and not ask them to search the guidelines again as each reviewer would like to do justice to an article without much waste of his/ her precious time. Please revert if any further information is required. This is a good article.I do not have any conflict of interest whatsoever with the authors or with any equipment manufacturer. Thank you once again for the opportunity of reviewing this aricle.Best wishes

Author Response

(The authors gave the same response as above.)
